# Implicit Curriculum in Procgen Made Explicit

**Zhenxiong Tan**[*]
National University of Singapore
zhenxiong@u.nus.edu

**Kaixin Wang**[*†]
National University of Singapore
kaixin96.wang@gmail.com

**Xinchao Wang**[‡]
National University of Singapore
xinchao@nus.edu.sg

## Abstract

Procedurally generated environments such as Procgen Benchmark provide a testbed for evaluating the agent's ability to robustly learn a relevant skill, by situating the agent in ever-changing levels. The diverse levels associated with varying contexts are naturally connected to curriculum learning. Existing works mainly focus on arranging the levels to explicitly form a curriculum. In this work, we take a close look at the learning process itself under the multi-level training in Procgen. Interestingly, the learning process exhibits a gradual shift from easy contexts to hard contexts, suggesting an implicit curriculum in multi-level training. Our analysis is made possible through *C-Procgen*, a benchmark we build upon Procgen that enables explicit control of the contexts. We believe our findings will foster a deeper understanding of learning in diverse contexts, and our benchmark will benefit future research in curriculum reinforcement learning.

## 1 Introduction

Training deep reinforcement learning agents in singleton environments is often susceptible to over-fitting (Zhang et al., 2018a,b; Song et al., 2019; Cobbe et al., 2019, 2020). Recently, procedurally generated environments, which take into account the agent's generalization ability, have received increasing attention (Justesen et al., 2018; Risi and Togelius, 2020; Cobbe et al., 2020). In those environments, the agent faces an entirely new level in each episode (*e.g.*, different maze layouts). Succeeding in such ever-changing levels demands that the agent learn robust policies. Compared to singleton environments, the diverse levels in procedurally generated environments pose new challenges for reinforcement learning training, calling for a better understanding of how learning progresses in this multi-level setting.

In procedurally generated environments, each level is associated with some environment parameters (Dennis et al., 2020) that vary across levels, *e.g.*, the agent's max speed, the size of the maze, or the number of obstacles. We refer to these various combinations of environment parameters as the contexts of the environment. The varying contexts give rise to levels of varying difficulties, suggesting a possible connection to curriculum learning (Bengio et al., 2009). While some previous works explore this connection, they mainly focus on arranging the order of levels to form a curriculum (Jiang et al., 2021b), or adaptively generating new levels to create a curriculum (Wang et al., 2019, 2020; Dennis et al., 2020; Jiang et al., 2021a; Parker-Holder et al., 2022). The learning process itself in the face of diverse levels is less investigated.

---

[*]Equal Contributions

[†]Currently in Microsoft Research Asia, work done during his time in NUS.

[‡]Corresponding Author.

38th Conference on Neural Information Processing Systems (NeurIPS 2024).

To bridge this gap, we take a closer look at how reinforcement learning progresses under multi-level training in Procgen. In particular, we do not consider any form of level prioritization, *i.e.*, the levels are sampled uniformly as in the standard setting of Procgen. To facilitate our study, we build a new benchmark, *C-Procgen*, which provides explicit access and control over the environment parameters. We then investigate how some metrics (*e.g.*, the agent's reward and the loss) under different contexts evolve as the learning progresses. In addition, we inspect how learning will be affected when some contexts are masked out or additional contexts are included.

We find that the learning process exhibits a gradual shift from easy contexts to hard contexts, despite the absence of explicit curriculum (*e.g.*, level prioritization). The return and the loss concentrate on easy tasks in the early stage of training, and gradually shift towards hard tasks as the training goes on. In addition, we observe a mismatch between the loss and the number of samples across different contexts in the middle stage of training. Specifically, a large portion of samples collected by the agent are from hard contexts but the loss is concentrated in easy tasks. This implies that a considerable number of samples might be wasted and contribute little to updating the policy. Moreover, we find that this mismatch has a potential connection to how much performance gain can be achieved from modifying the sample distribution by prioritizing levels (Jiang et al., 2021b).

When the training contexts are partially masked out (easy, medium, or hard), we observe mixed results: in some games, the excluded contexts are essential for the agent to form an implicit curriculum, and masking them out will impact the performance; in other games, the remaining contexts are still able to provide useful information for the agent to learn transferrable policies. Besides, we find that expanding the set of training contexts helps expedite the learning shift from easy to hard contexts.

Our C-Procgen benchmark exposes the environment parameters in the black-box context generation process in Procgen. It preserves the advantages of the Procgen benchmark, such as being comprehensive and challenging, while greatly expanding its use by adding explicit control of the contexts. C-Procgen can be useful in curriculum learning and contextual reinforcement learning.

In summary, our paper makes the following contributions:

- We build C-Procgen, a benchmark that enhances Procgen with accessible and controllable environment parameters, which can be of interest to curriculum learning and other areas.

- Our work offers insights into the learning dynamics under the multi-context training in Procgen, and makes several interesting observations.

- We reveal that an implicit curriculum occurs in the multi-context training in Procgen despite that the contexts are uniformly sampled.

## 2 Related works

### 2.1 Procedurally generated environments

Recently, procedural content generation has been adopted to design RL environments that consider the agent's ability to generalize as a central component of success. In procedurally generated environments, each episode begins with a new context created from a generation process, which is analogous to a "level" in video games. Across different episodes, the context is varied in certain task-relevant (*e.g.*, number of obstacles) or task-irrelevant (*e.g.*, background image) aspects. One popular procedurally generated environment suite is the Procgen benchmark (Cobbe et al., 2020), which consists of 16 challenging Atari-like video games. However, in Procgen, multiple choices during the procedural generation process are governed by a single random seed. The underlying parameters that control the context generation are not accessible and visible to researchers. In other words, the context is implicitly determined by a black-box process. As pointed out by (Kirk et al., 2023), the evaluation protocols supported by such environments are limited to varying the size of the context set. Our work empowers Procgen by making many factors of variation visible and controllable, allowing researchers to explicitly configure the environment parameters.

Additionally, several works like Minigrid (Chevalier-Boisvert et al., 2018) and Minihack (Samvelyan et al., 2021) offer flexible frameworks for creating customizable reinforcement learning environments with adjustable parameters to generate different contexts. However, their environments are lightweight and lack convenient APIs for controlling environment parameters.

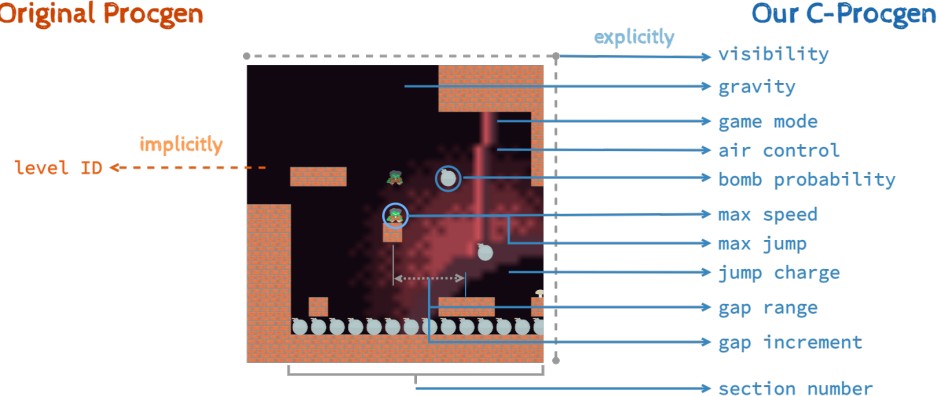

Figure 1: An illustrative example comparing the original Procgen and our C-Procgen.

Closely related to our work is the CARL benchmark (Benjamins et al., 2021), which collects environments from several domains and makes their contexts configurable. While CARL mostly focuses on control tasks (including classic control environments from OpenAI Gym (Brockman et al., 2016) and physical simulations), our C-Procgen aims to extend the widely used Procgen benchmark.

## 2.2  Curriculum learning

Curriculum learning (Bengio et al., 2009) is a general strategy in machine learning that organizes training examples or tasks in order to boost learning efficiency. The varying contexts in Procgen pose challenges at different levels of difficulty, which is naturally connected to curriculum learning. Jiang et al. (2021b) propose to prioritize the contexts that have higher estimated learning potential when sampling new contexts for the next training levels. This induces an emergent curriculum of increasingly difficult levels. Another line of work aims to automatically generate novel contexts that adapt to the agent's evolving ability (Wang et al., 2019, 2020; Dennis et al., 2020; Jiang et al., 2021a; Parker-Holder et al., 2022). While these works focus on manipulating the distribution of contexts to explicitly form a curriculum, our paper reveals that even when the contexts are uniformly sampled, the agent's learning process still implicitly follows a curriculum.

The prior works mentioned above can also be viewed as a form of automatic curriculum learning (Portelas et al., 2020). In this regard, our work is also closely related to the TeachMyAgent benchmark (Romac et al., 2021). TeachMyAgent extends the Bipedal Walker environment from OpenAI Gym (Brockman et al., 2016) to parametric ones that enable controllable procedural generation, providing a testbed for benchmarking automatic curriculum learning algorithms. Similarly, our work aims to make the more challenging Procgen benchmark configurable.

## 3  C-Procgen: Controllable Contextual Procgen

In this work, we first build a new benchmark that enhances the challenging Procgen suite aiming to investigate how learning progresses under the multi-context setting. Specifically, we carefully refactor the source code of Procgen and expose over 100 environment parameters that determine the context generation, giving users more control and insight into the previously opaque generation process. The resulting benchmark is **C-Procgen** [4] , which augments the original Procgen with **C**ontrollable **C**ontextual environment parameters and is well-suited for **C**urriculum reinforcement learning research. Throughout this paper, we use the term "environment parameter" (Dennis et al., 2020) to refer to any configurable factor that can vary across episodes (e.g., the agent's max speed, maze size, or number of obstacles). The term "context" refers to a specific configuration determined by the combination of all environment parameters.

As illustrated in Figure 1, the original Procgen only provides a level ID to implicitly control the generation of context. In contrast, our C-Procgen allows users to directly control the context

---

[4]The source code of C-Procgen can be found on GitHub: https://github.com/zxtan98/CProcgen

via explicit configuration of the environment parameters. Thus, C-Procgen can serve as a high-quality testbed for research in the area of Contextual Reinforcement Learning (Hallak et al., 2015). Furthermore, C-Procgen enables users to integrate the context information into the learning algorithms, which can be useful in Automatic Curriculum Learning (Portelas et al., 2020).

In addition, C-Procgen also provides several engineering enhancements for improved usability:

- Unlike the original Procgen, which assigns the same configuration to all environments during the initialization of vectorized environments, C-Procgen allows distinct contexts to be assigned to each environment. This flexibility helps expose the algorithm to more diverse game contexts during training.

- Users can dynamically modify the context of each environment between episodes without creating a new instance. This feature is particularly beneficial for curriculum reinforcement learning, where environments need dynamically changing contexts throughout the learning process.

- C-Procgen offers an interface that lets users track detailed context information during the learning process, enhancing transparency and facilitating better analysis.

In summary, we would like to highlight that C-Procgen enhances Procgen with controllable environment parameters while the introduced overhead during simulation is negligible. In the following sections, we will take advantage of the features provided by C-Procgen to conduct a detailed analysis of how the learning dynamics evolve across contexts in Procgen.

## 4    Learning Dynamics in Procgen

In this section, we investigate the learning progress of Procgen under various contexts. For our experiments, we used C-Procgen, which faithfully simulates the same game logic and context distributions as the easy mode of the original Procgen benchmark. By leveraging the flexible context control features of C-Procgen, we recorded key metrics such as loss, entropy, episode length, average score, and the number of samples for each context. This approach provides a more detailed view of learning progress across different contexts.

Specifically, we select nine environments from Procgen due to their episodic contexts, which change with each game reset, resulting in unique configurations for each playthrough. We utilize the Proximal Policy Optimization (PPO) algorithm (Schulman et al., 2017) in our reinforcement learning experiments. For each of these selected environments, we perform five individual runs, each encompassing 25 million steps, to ensure a comprehensive and robust analysis. Notably, the experiment with C-Procgen allows for the generation of diverse environments under all possible circumstances, following the sample efficiency experimental protocols of the original Procgen benchmark.

### 4.1    A Shift from Easy to Hard Contexts

To gain deeper insights into the learning process in C-Procgen, we first focus on the game `Ninja`. In the game `Ninja`, the agent navigates through a field of suspended platforms while avoiding bombs. This environment contains six unique contexts, each determined by different environment parameters including `section_num`, `gap_range` and `bomb_prob`. For instance, the number of platforms in the game, determined by the contextual factor `section_num`, significantly impacts the gameplay. Moreover, the environment parameter `gap_range` determines the distance between the two platforms. Smaller values correspond to smaller gaps, thus implying easier transitions for the agent. Furthermore, a higher `bomb_prob` escalates the likelihood of encountering bombs. For instance, Figure 2 presents heat maps that visualize some metrics of the learning process across different contexts within the game `Ninja`, where each grid indicates a unique context. [5] Take, for example, the grid at coordinates $(2, 1)$; it corresponds to a context where the game features two platforms, and the `gap_range` and `bomb_prob` are set to their respective minimum values.

---

[5]Due to their direct correlation, the environment parameters `gap_range` and `bomb_prob` are combined on the same axis for simplification. Specifically, as `gap_range` increases, `bomb_prob` also increases. They are correlated in the following three combinations: when `gap_range` is `[0]`, `bomb_prob` is 0; when `gap_range` is `[0, 1]`, `bomb_prob` is 0.25; and when `gap_range` is `[0, 1, 2]`, `bomb_prob` is 0.5.

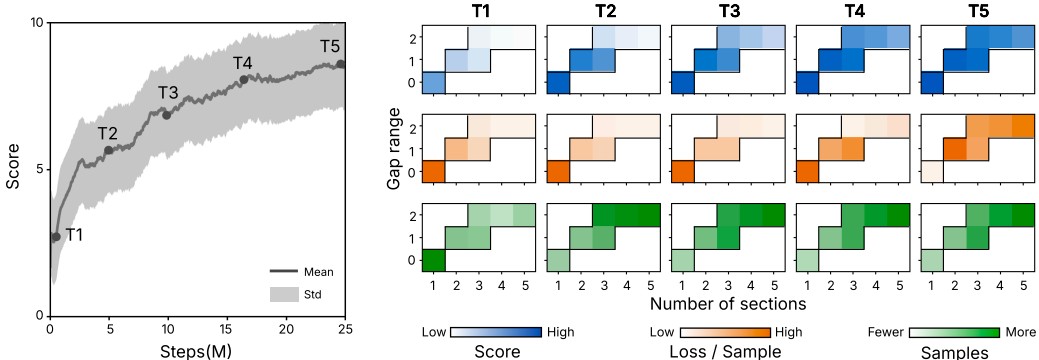

Figure 2: Dissecting the learning dynamics in Procgen with game `Ninja` as an example. **Left**: The training curve, where we mark five stages during training (T1-T5). **Right**: Heatmaps of three metrics at stage T1-T5, including average score (top row), loss per sample (middle row), and the number of samples (bottom row) across different contexts.

**Average Score.** A notable phenomenon is observed in the *Average Score* for `Ninja` (depicted in blue heat maps, see Figure 2). The average score, which measures the agent's performance, varies significantly across different contexts. Notably, as early as stage $T_2$, the agent demonstrated high scoring capability in the context $(1, 0)$, while in other more challenging contexts such as $(5, 2)$, high scores were only achieved at a later stage as $T_5$. This led us to categorize the contexts into two types: "Easy contexts", where high scores were obtained early in training, and "Hard contexts", where high scores were achieved much later. This progression of mastering different contexts effectively forms a curriculum, subtly guiding the agent's learning process.

**Loss per Sample.** Another important insight concerns the *Loss per Sample* of `Ninja` (depicted in orange color heat maps, see Figure 2 [6]. This metric represents the average absolute value of the sample-wise loss [7] generated at each algorithmic step within a specific context.

More specifically, throughout the learning process, the focus of the loss per sample metric appears to shift progressively from simpler contexts to more challenging ones, potentially reflecting the order in which the agent learns to master different scenarios. Notably, this metric tends to concentrate on contexts that pose a moderate level of difficulty for the agent – those that are neither too easy nor too difficult. This phenomenon can be understood as follows: in contexts where the agent is either highly proficient or completely unfamiliar, it can make more accurate value judgments, resulting in fewer losses. However, in moderately challenging contexts, the agent encounters more variability in rewards, leading to errors in value estimation and, consequently, a higher number of loss signals to drive its updates. This observation aligns with the core principle of curriculum learning, which stresses the importance of guiding the agent through tasks that are neither too easy nor too difficult.

**Samples Distribution.** An analysis of the *Samples Distribution* (depicted in green color heat maps, see Figure 2 reveals insights about the distribution of samples across different contexts within a single iteration. While the probability of sampling each context remains constant during environment resets and does not change over time, the length of the episode within each context is not constant, leading to variation in the samples across contexts. Aside from the earliest stage at $T_1$, it is observed from the heat map that for the majority of the learning process within the game `Ninja`, samples tend to concentrate within the hard contexts.

Figure 3 expands the scope of observation beyond the `Ninja` environment to encompass a wider range of environments. We found that the observations made in the `Ninja` environment can be generalized to other environments of the Procgen benchmark.

---

[6]The value ranges in the heatmaps of Figure 2 and Figure 3 have been normalized. The range for the score is based on the maximum and minimum values achieved by the agent over the entire learning process, while the ranges for loss per sample and the number of samples are based on the maximum and minimum values at each specific time point.

[7]The loss from PPO (Schulman et al., 2017) is in a weighted sum of three terms: $l_{\text{policy}}$, $l_{\text{value}}$ and $l_{\text{entropy}}$.

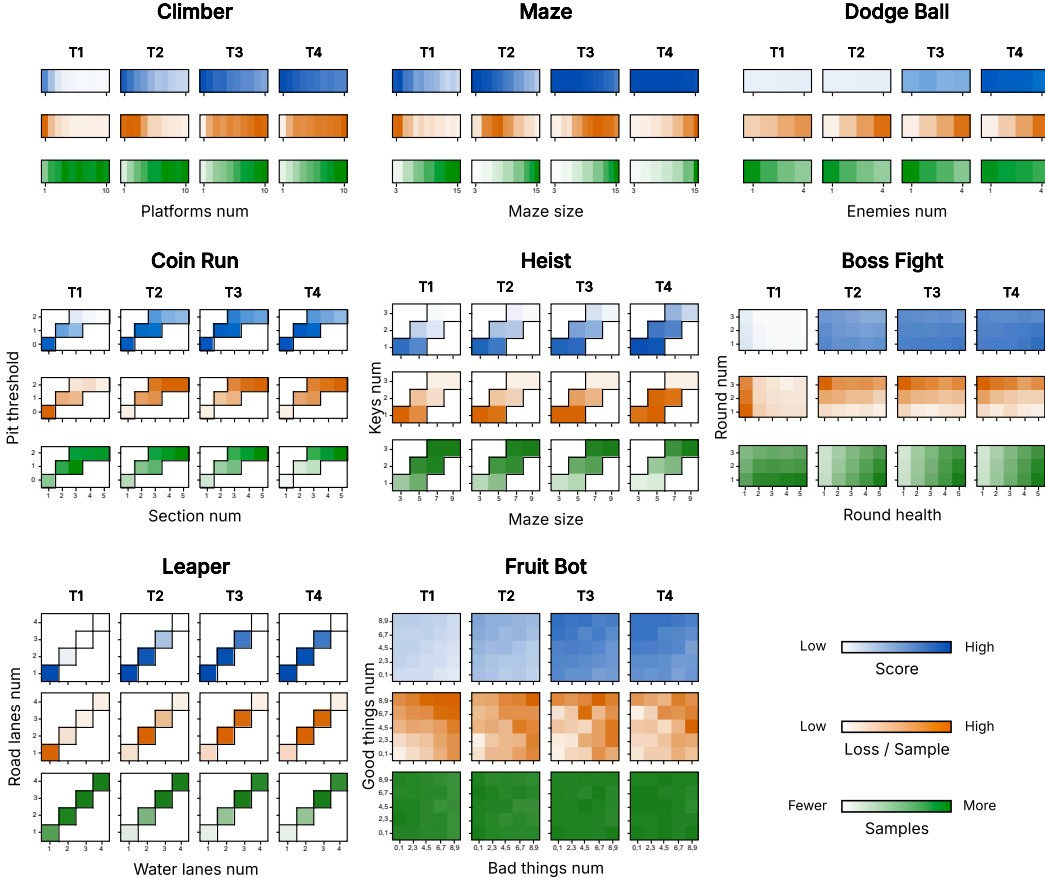

Figure 3: Score, loss per sample, and sample num across different environments and contexts. Each subplot represents a different environment, with heatmaps depicting the metric values at different time points and contexts.

## 4.2 A Mismatch between Loss and Sample Distribution

The metric "Loss per sample" indicates whether the agent has learned new content in a specific context to a certain extent. As previously discussed, the agent's focus tends to gravitate toward contexts of moderate difficulty during learning. However, we observe an inconsistency: the contexts on which the agent focuses do not match the distribution of samples in the Procgen environments. This discrepancy suggests that a significant portion of the samples may not be providing sufficient loss signals to effectively guide the agent's learning. Particularly in the early stages of the process, easy contexts often promote the agent's initial progress, yet a majority of the samples the agent encounters come from hard contexts, which yield limited learning.

To gain a deeper understanding of this inconsistency, we introduce a measure called "Loss Production Efficiency" that captures the effectiveness of sample distribution in generating loss signals at the $i^{th}$ iteration throughout the learning process in Procgen. The Loss Production Efficiency is defined as the ratio between the loss generated by the current sample distribution across different contexts and the maximum achievable loss:

$$E_i = \frac{\sum_{c \in \mathcal{C}} p_i^c \cdot l_i^c}{\max_{c \in \mathcal{C}} l_i^c} \tag{1}$$

In this equation, $E_i$ represents the Loss Production Efficiency (LPE) for a specific context $c$ from context space $\mathcal{C}$ at the $i^{th}$ iteration. Here, $l_i^c$ refers to the loss per sample for context $c$ at iteration $i$, and $p_i^c$ denotes the proportion of samples for context $c$ out of the total samples in the same iteration. The

LPE for the entire learning process is represented by the average of the Loss Production Efficiencies across all iterations, given by $E = \frac{1}{N} \sum_i^N E_i$, where $N$ is the total number of iterations.

A notable observation reveals a potential correlation between the LPE and the performance improvement offered by the Prioritized Level Replay algorithm (Jiang et al., 2021b) on Procgen games. The performance improvement is defined as $\Delta score = \frac{score_{\text{PLR}} - score_{\text{PPO}}}{score_{\text{PPO}}}$, where $score_{\text{PLR}}$ and $score_{\text{PPO}}$ are the final scores with and without PLR, respectively. The Prioritized Level Replay algorithm aims to enhance the learning process by augmenting data from samples generated by high-loss levels. Interestingly, as illustrated in Figure 4, it appears that environments characterized by lower LPE tend to exhibit more substantial performance gains when the Prioritized Level Replay algorithm is employed. This could be attributed to the algorithm's ability to modify the sample distribution and increase LPE.

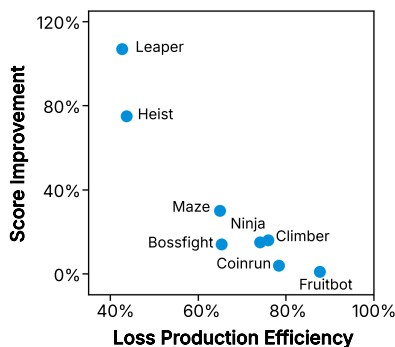

Figure 4: Loss Production Efficiency vs. Score Improvement with Prioritized Level Replay in Procgen Games

Based on the insights gained from the above findings, it becomes apparent that LPE plays a crucial role in the learning speed within an environment. When designing environments, optimizing factors such as the distribution of contexts, maximum episode length, and conditions for terminating episodes can significantly impact LPE. By carefully manipulating these factors, it is possible to create environments that promote higher LPE, resulting in accelerated learning processes and improved agent performance.

## 5 Investigating Learning Dynamics under Context Reconfiguration

To investigate the influence of different training contexts on the reinforcement learning process, we reconfigure the context setting of games using C-Procgen. Firstly, based on previous findings of loss per sample across different contexts, we grouped contexts into three categories. We then created game settings excluding these context groups, to test their necessity to learning progress. Secondly, we add new contexts into the games, aiming to decrease discontinuity between existing contexts. We employed the Proximal Policy Optimization (PPO) algorithm, conducting 5 runs for each setting, consistent with the procedures in Section 4.

### 5.1 Partially Masking the Training Contexts

The experiments demonstrate the distinct roles of contexts in different games. As depicted in Figure 5, we present the final average score and loss per sample heat maps for nine games when trained without certain contexts.

Overall, certain contexts appear to be crucial for the learning process in the game. For instance, in setting 2 of `Coinrun`, removing key contexts causes the learning process to stall, preventing the agent from progressing through the game's various stages. Similar patterns are observed in setting 1 of `Heist` and setting 2 of `Ninja`. Moreover, a generalization phenomenon was observed across multiple environments. In many cases, the agent was able to perform well even without training in certain contexts. Interestingly, the overall performance of the agent improved in some cases when certain contexts were excluded, such as in setting 3 of `Climber`, Setting 3 of `Coinrun`, and both Setting 2 and Setting 3 of `Heist`. One possible explanation for this improvement is that the excluded contexts were particularly difficult. By removing these challenging contexts, the agent may have experienced increased learning progress efficiency (LPE), which in turn enhanced its overall performance.

In summary, our observations indicate that certain contexts play a crucial transitional role in the learning process. However, the benefit of having a larger number or wider variety of contexts is not always straightforward.

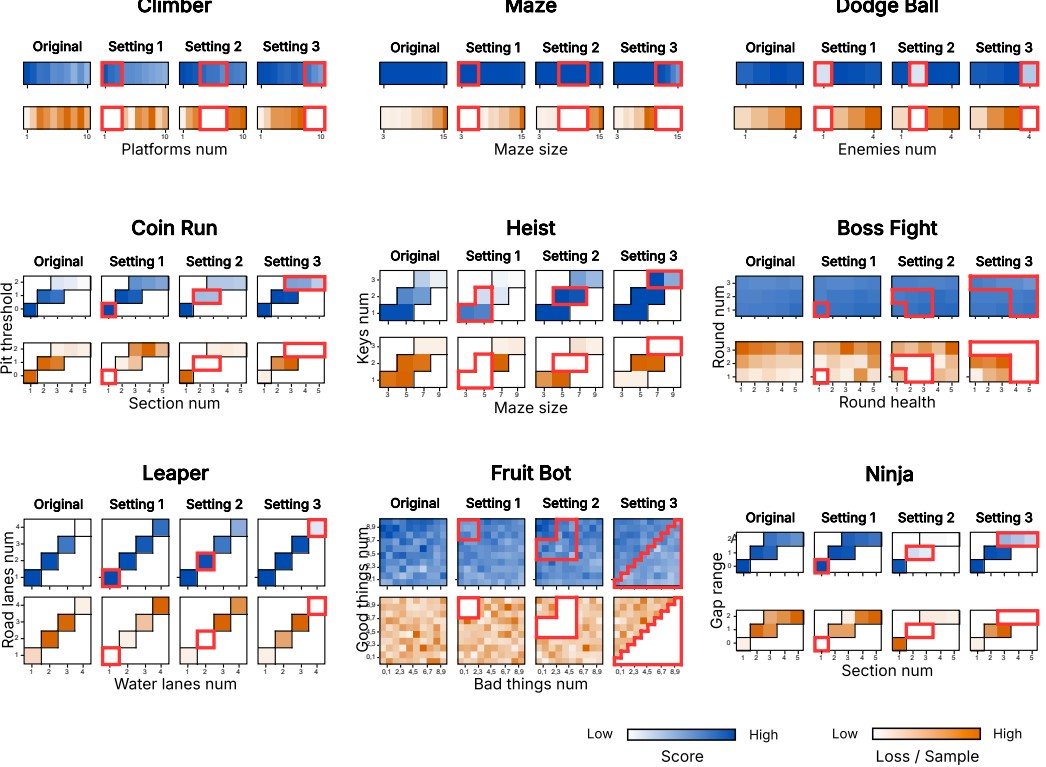

Figure 5: Learning outcomes under various context settings. The leftmost panel for each game depicts the final average score and loss per sample under the original context setting after 25M training steps. Subsequent panels show the metrics under reconfigured context settings. Regions enclosed in red represent the masked contexts, where average score testing was still performed.

## 5.2 Expanding the Training Contexts

In this section, we focus on the game `Leaper` as a case study. We aim to provide a detailed exploration of how the introduction of new contexts in the original Procgen benchmark, can influence the learning progress, particularly by analyzing the changing distribution of loss during the learning process.

The game environment of `Leaper` requires the agent to avoid road vehicles and make use of the logs on the river to reach the top of the map. Within this environment, there are two environment parameters: the number of road lanes (`road_lanes_num`) and the number of water lanes (`water_lanes_num`). Under the original context setting of `Leaper`, only four contexts exist: $(0, 0)$, $(1, 1)$, $(2, 2)$, and $(3, 3)$. This indicates that the game will have an equal number of water lanes and road lanes, with each context having an equal chance of being sampled at game initialization. In addition to the original context setting of `Leaper`, we have introduced two new context settings. In Setting 1, we have six new contexts to enhance the connectivity among the original context setting. In Setting 2, we permit both `water_lanes_num` and `road_lanes_num` to take any value within the set $\{0, 1, 2, 3\}$. In both Setting 1 and Setting 2, each context has an equal probability of being sampled.

With the original context setting, as observed from the stacked area chart in Figure 6, the loss during the early stages of learning is primarily focused on context 0, followed by context 1. The transition of loss focus to context 2, however, is relatively slow, and by the end of training, the loss has not yet concentrated on context 3. In contrast, in Setting 1, due to the existence of more diverse contexts and stronger connectivity among them, the proportion of loss generated by contexts 1 and 2 starts to shift towards context 3 as early as around the 2 million steps. Comparing the results of the original setting and Setting 1, it is suggested that the expanded contexts serve as stepping stones, aiding in the agent's learning and adaptation across different contexts. When comparing Settings 1 and 2, Setting 2 also

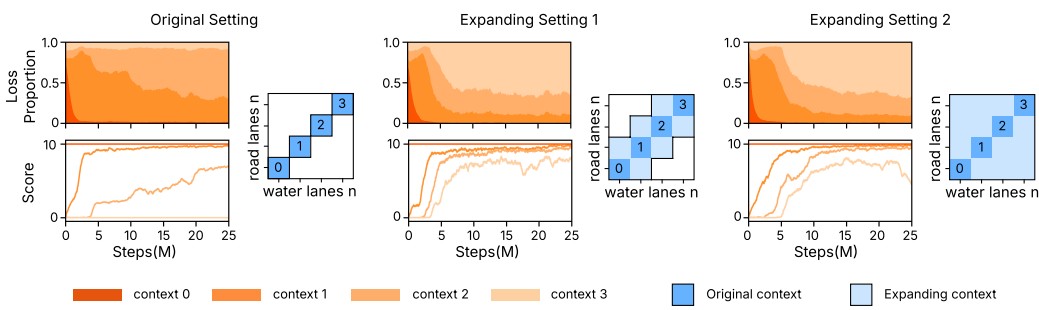

Figure 6: Loss Proportion and Score curve in different expanded context settings

accelerates learning, but its effects appear less significant than those in Setting 1. This observation is consistent with the trends noted in Section 5.2.

Moreover, upon observing the score curves and loss proportion of these three settings (see Figure 6), it is notable that the increase in the agent's loss proportion corresponds to the time when the agent's average score in that context begins to rapidly grow. Conversely, when the agent's loss proportion begins to decline in a context, it suggests that the agent has already become proficient in that context. Consistent with the findings in Section 4, when the loss begins to shift, the agent has indeed already mastered the respective context.

# 6 Conclusions

Procgen is a widely used procedurally generated environment suite that has varying contexts across episodes. While previous works pay most attention to explicitly creating a curriculum of contexts, we seek to investigate the learning process itself under the multi-level training in Procgen. To this end, we build C-Procgen, which enhances Procgen with explicit access and control of context parameters. With C-Procgen, we analyze the learning dynamics in Procgen across contexts and reveal that an implicit curriculum happens as the learning proceeds. Our C-Procgen can also find its use in other curriculum learning research. In the future, we will explore if this implicit curriculum connects to the number of contexts and how we can take advantage of it to boost performance.

# 7 Limitation

We did not provide specific evaluation protocols for C-Procgen. Even though Procgen offers a comprehensive set of games, we believe there are many other types of games that it doesn't cover. While we conducted several experiments, notably those involving modified context distributions, we did not explore dynamic context distributions — where the context distribution evolves with the learning progress. We believe that studies in this direction hold significant value for curriculum learning. We plan to study this deeper in our future work.

## Acknowledgement

This work is supported by the Advanced Research and Technology Innovation Centre (ARTIC), the National University of Singapore under Grant (project number: A-0005947-21-00, project reference: ECT-RP2), and the Singapore Ministry of Education Academic Research Fund Tier 1 (WBS: A-0009440-01-00).

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

# A    Experiment Details

We use the PyTorch codebase from Raileanu and Fergus (2021) to train PPO Schulman et al. (2017) agents on Procgen games. The agent is parameterized by the IMPALA network architecture Espeholt et al. (2018). Unless otherwise stated, we follow the hyperparameters used in Cobbe et al. (2020) for the easy mode of Procgen, as summarized in Table 1.

Similar to the original Procgen, our C-Procgen still enjoys high simulating speed. On a server with 2 Intel Xeon CPU cores and 56GB RAM, the FPS of Procgen is around 750 and the FPS of C-Procgen is around 710. Regarding the training cost, training a PPO agent on an NVIDIA T4 GPU for 25M steps takes approximately 2.5 3 hours.

Table 2 provides information on the episodic contexts used in our experiments in Sections 4 and 5, including their descriptions and ranges of values.

| HYPERPARAMETER | VALUE |
| --- | --- |
| $\gamma$ | 0.999 |
| $\lambda$ | 0.95 |
| # timesteps per rollout | 256 |
| # epochs per rollout | 3 |
| # minibatches per epoch | 8 |
| entropy bonus | 0.01 |
| clip range | 0.2 |
| reward normalization | no |
| learning rate | 5e-4 |
| # workers | 1 |
| # environments per worker | 64 |
| # total timesteps | 25M |
| optimizer | Adam |
| LSTM | no |
| frame stack | no |

Table 1: Hyperparameters and their values

| GAME | ENVIRONMENT PARAMETER | DESCRIPTION | VALUES |
|---|---|---|---|
| bossfight | Number of rounds | Determines the health or durability of each round (boss starship). Higher values indicate a more resilient boss that requires more damage to be defeated. | $[1, 2, 3]$ |
| | Round health | Specifies the number of rounds in the game. Each round represents an encounter with the boss starship. | $[1, 2, 3, 4, 5]$ |
| climber | Number of platforms | Specifies the number of platforms in the game. Each platform represents a step for the player to climb. | $[1, 2, \cdots, 10]$ |
| coinrun | Number of sections | Determines the number of sections present in each game. | $[0, 1, 2, 3, 4]$ |
| | Pit threshold | Affects the generation of pits in the game. When the value is smaller, there is a higher probability of pits appearing in the game. Conversely, a larger value reduces the likelihood of pits. This environment parameter interacts with several other environment parameters, which are further detailed in the code documentation under the "context details" section. | $[0, 1, 2]$ |
| dodgeball | Number of enemies | Specifies the number of enemies present in the game. The player must dodge the balls thrown by these enemies and eliminate them to progress. | $[1, 2, 3, 4]$ |
| fruitbot | Number of good things | Specifies the number of fruits that the player can collect along the way. Collecting a piece of fruit rewards the player with a positive reward. | $[0, 1, \cdots, 9]$ |
| | Number of bad things | Specifies the number of non-fruit objects that the player must avoid. Mistakenly collecting a non-fruit object results in a larger negative reward. | $[0, 1, \cdots, 9]$ |
| heist | Number of keys | Specifies the total number of keys required to unlock the locks and successfully complete the heist. The player must collect these keys scattered throughout the game. | $[1, 2, 3]$ |
| | Size of maze | Determines the size of the maze layout in which the heist takes place. The maze is generated using Kruskal's algorithm and serves as the environment for the player to navigate and find the hidden gem. | $[3, 5, 7, 9]$ |
| leaper | Number of road lanes | Specifies the number of lanes in which cars move. The player must cross these lanes to reach the finish line and earn a reward. | $[0, 1, 2, 3]$ |
| | Number of water lanes | Specifies the number of lanes with logs on a river. | $[0, 1, 2, 3]$ |
| maze | Size of maze | Determines the size of the maze layout. | $[3, 4, \cdots, 15]$ |
| ninja | Number of sections | Specifies the number of narrow ledges that the player, a ninja, must jump across. | $[0, 1, 2, 3, 4]$ |
| | Gap range | Controls the range of gaps between the ledges that the ninja must traverse. | $[0, 1, 2]$ |

Table 2: Descriptions of the contexts used in our experiments.

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
