# OpenReview forum: "Implicit Curriculum in Procgen Made Explicit"
_NeurIPS.cc/2024/Conference — NeurIPS 2024 spotlight_

### Official Review · Reviewer_2ABJ · 2024-06-23

**Soundness:** 3
**Presentation:** 2
**Contribution:** 3
**Rating:** 6
**Confidence:** 5

**Summary:**

This paper introduces C-Procgen, an extension to the popular Procgen benchmark that includes many improvements over the original, most notably by adding "context parameters" that can control specific features of the procedural content generation. They thoroughly study the performance of agents in different contexts and identify an implicit curriculum in which agents spend more time and experience higher loss values in progressively harder contexts as training progresses. They also define a new metric for measuring loss signal efficiency and identify a correlation between this metric and the performance of Prioritized Level Replay, a popular curriculum learning method. Finally, the paper investigates how agent performance is affected by masking out specific contexts from the training set.

**Strengths:**

Overall, the paper provides several substantial contributions to reinforcement learning and curriculum learning. Procgen is one of the most popular benchmarks for evaluating generalization in RL, and this work provides valuable new features and analyses for this benchmark. The context parameters introduced by C-Procgen would allow researchers to use Procgen to study more varied curriculum learning methods. The authors seem to have this specific use case in mind as they implemented several important features for curriculum learning, such as allowing environments to be efficiently configured during a reset, and allowing each environment in the vector environment to be individually configured.

The analysis of individual procgen environments also identifies which environments and which components of those environments are challenging for RL agents to learn. It can be unintuitive to identify what RL agents may struggle to learn, so these findings may be valuable for interpreting future results using Procgen. By sorting contexts according to difficulty, it is possible to create an explicit curriculum to train agents with, as the authors allude to. Many automatic curriculum methods aim to automatically sort tasks by difficulty, so this may serve as a strong baseline for evaluating those methods.

Overall, the methodology, presentation of results, and analyses are all sound. The figures are clear and easy to interpret. The claims in the paper seem to match to experimental results presented in the figures, and the authors do a good job highlighting interesting findings aside from their main claims.

**Weaknesses:**

This paper has several weaknesses, mostly regarding clarity and comparison to relevant literature. It would have been nice to see some comparison to other environments that are typically used in curriculum learning and generalization research, such as Minigrid [1] and Minihack [2], which provide similarly configurable gridworld environments. The authors do compare their work to CARL, but miss some popular benchmarks. A more glaring gap is that this work does not reference Unsupervised Environment Design [3], which introduces a underspecified POMDP formulation that extends POMDPs to have configurable free parameters, exactly like the context parameters used in C-Procgen.



Overall the writing quality of the paper is relatively paper, particularly in the introduction. There are numerous grammatical errors, just to highlight a few:

* Line 31: "which provides explicit access and *control to* the context parameters" should probably say *control over*.
* Line 32:  "The return and loss *concentrate in* easy tasks" should say *concentrate on.*
* Line 40: "samples collected by the agent is from hard contexts" should say *are from*.
* Line 207: "*Level Replay*" should say *Prioritized Level Replay*.

Aside from these examples, the phrasing is somewhat awkward in many places. This is exacerbated by the fact that definitions of key terms do not appear until section 3 if at all. "Context parameters" are referenced multiple times before being defined on Line 102. The authors often reference the "learning process" in a vague way. For example, on Line 26 they make the claim that in previous work on curriculum learning "The learning process itself in the face of diverse levels is less investigated." Clearly those works utilize and study learning processes, so it is unclear what comparison the authors are trying to make. It is possible that drawing on the terminology from unsupervised environment design may help to clarify what context and contextual parameters are.



I also have a few concerns regarding soundness and novelty:

* Using loss on its own as a metric is usually not well justified for policy gradient methods, but the authors show that their loss production efficiency metric is negatively correlated with score improvement from using PLR. However, the paper does not explain how score improvement in Figure 4 is calculated.
* It's not clear from the paper whether they are using PPO's value loss, policy loss, or both.
* The finding that agents collect more samples from harder contexts as training progresses seems obvious by definition. If we have a reward function that is correlated with survival, and episodes end upon death, then as the agent learns to maximize reward, it is learning to survive for longer and therefore collect more samples from challenging contexts. The opposite would imply that the agent is learning to minimize reward. This is corroborated by the figures in which the score for every context increases during training.
* It is unsurprising that more challenging contexts results in higher loss values, and its likely that this has been observed in other environments in previous work.
* Despite training agents for many diverse contexts, this work does not appear to include training curves or scores for those agents. Training curves can be a valuable reference for researchers looking to use new benchmarks. It also impossible to tell from the paper whether C-Procgen and Procgen agents perform similarly.



[1] Chevalier-Boisvert, Maxime, et al. "Minigrid & miniworld: Modular & customizable reinforcement learning environments for goal-oriented tasks." *Advances in Neural Information Processing Systems* 36 (2024).

[2] Samvelyan, Mikayel, et al. "Minihack the planet: A sandbox for open-ended reinforcement learning research." *arXiv preprint arXiv:2109.13202* (2021).

[3] Dennis, Michael, et al. "Emergent complexity and zero-shot transfer via unsupervised environment design." *Advances in neural information processing systems* 33 (2020): 13049-13061.

**Questions:**

* On line 142, how does a continuous value `bomb_prob` produce only 6 unique contexts?
* On line 149, if `gap_range` and `bomb_prob` can be individually configured, how are they correlated?
* Does this work use policy loss, value loss, or both?
* Is it still possible to seed a C-Procgen environment with the original procedural content generation? This may be important for comparison to prior work.
* How was the performance improvement for PLR calculated or evaluated?
* Have the authors investigated whether previous work has identified higher loss values in challenging states or environments?

**Limitations:**

The authors list several limitations in their paper, namely that they do not define new evaluation protocols for C-Procgen, and that they do not explore curriculum learning. Curriculum learning itself seems outside the scope of this paper, though it would make a nice addition. It does seem problematic that the work provides no training curves or evaluation protocols for future work. Without them, future work may decide on different protocols, making them harder to compare and limiting reproducibility.



The paper also does not discuss whether their results could generalize to other learning algorithms aside from PPO. It is common for work in this space to only use PPO, but that limitation should be stated in the paper. The findings in this paper are also largely Procgen specific and unlikely to generalize to other RL environments.

---

> ### Author Rebuttal · Authors · 2024-08-07
>
> We highly appreciate the feedback from the reviewer-2ABJ and will address each point in detail below.
>
> ---
> **W1. Issues Regarding to Literature Comparison**
>
> We appreciate the highlighting of Minigrid and Minihack as notable works similar to our C-Procgen, and regret the oversight of not mentioning them in our initial submission. Additionally, we acknowledge the omission of '*Unsupervised Environment Design*' in our submission, despite its relevance and prominence in the field.
>
> 1. We agree that Minigrid and Minihack are widely used in curriculum learning due to their customizable environments. While Procgen offers a broader variety of challenges, it lacks the customizable features essential for detailed control, which we have incorporated into C-Procgen.
>
> 2. After reviewing '*Unsupervised Environment Design*', we recognized its overlap with C-Procgen, especially in using configurable parameters for training environments. We will incorporate terms like 'underspecified environment' and 'free parameters' to clarify and deepen our discussion!
>
> ---
> **W2. Typos**
>
> We appreciate the detailed feedback on the typographical errors in our paper. We have thoroughly reviewed the manuscript and corrected these issues to provide a revision. Thanks to the reviewer for highlighting these points.
>
> ---
> **W3. Lack of Clarity and Definitions**
>
> We appreciate feedback from the reviewer on clarity.
> We will define key terms like 'context parameters' earlier in the text and refine our descriptions of the learning process to address these concerns more directly.
> And, we would love to align our terminologies with those used in '*Unsupervised Environment Design*' in our revision.
>
> ---
> **W4 & Q5. Lack of Metric Explanation**
>
> We apologize for the oversight and will address this in our revision.
> The score improvement which is defined as
> $\frac{score_{\text{PLR}} - score_{\text{PPO}}}{score_{\text{PPO}}}$.
> Here, $score_{\text{PLR}}$ and $score_{\text{PPO}}$ represent the scores achieved by the PLR and PPO algorithms respectively.
>
> ---
> **W5 & Q3. Loss Specification**
>
> We apologize for not specifying this clearly in the manuscript. The 'loss' referred to in the paper is the PPO Loss, which is composed in a weighted sum of three terms:
> $l_{policy}$,
> $l_{value}$,
> and $l_{entropy}$.
>
> ---
> **W6. Obviousness of Training Progression Findings**
>
> As the reviewer pointed out, it might seem unsurprising that agents collect more samples from harder contexts as they learn to prolong survival, especially when episodes end upon the agent's 'death.'
> However, this is not the only interpretation.
> - For example, in a maze scenario, an agent might accumulate many steps in challenging contexts not because it is effectively learning to navigate but because it cannot quickly exit the maze, potentially ending the episode with zero reward after expending many steps.
> - While initially this may not seem surprising, on reflection, the implication that an agent spends significant steps in unrewarding contexts is noteworthy.
>
> ---
> **W7.**
> > Reviewer: It is unsurprising that more challenging contexts results in higher loss values.
>
> We observed that the contexts resulting in higher loss values dynamically change throughout the learning process, rather than being consistently fixed on more challenging contexts.
> Interestingly, this phenomenon appears to be nearly universal across all environments tested.
>
> ---
> **W8.**
>
> Thanks for the suggestion! We have included learning curves with more detailed metrics information in the global response PDF attached. Upon comparison, we found that the performance of PPO on C-Procgen is largely consistent with its performance on Procgen.
>
> ---
> **Q1:**
>
> > Reviwer: On line 142, how does a continuous value bomb_prob produce only 6 unique contexts?
>
> Although `bomb_prob` can be a continuous value, in the game logic, it is restricted to three discrete values: `0`, `0.25`, and `0.5`. The six unique contexts are determined by these discrete values and the number of sections (num_section). Specifically, the contexts setting of `num_section` and `bomb_prob` are as follows: [1, 0], [2, 0.25], [3, 0.25], [3, 0.5], [4, 0.5], [5, 0.5],
>
> ---
> **Q2:**
>
> > Reviwer: On line 149, if gap_range and bomb_prob can be individually configured, how are they correlated?
>
> In the game logic, as `gap_range` increases, `bomb_prob` also increases. They are correlated in the following three combinations: when `gap_range` is [0], `bomb_prob` is 0; when `gap_range` is [0, 1], `bomb_prob` is 0.25; and when `gap_range` is [0, 1, 2], `bomb_prob` is 0.5.
>
> ---
> **Q4:**
>
> > Reviwer: Is it still possible to seed a C-Procgen environment with the original procedural content generation?
>
> Yes, it is indeed possible to replicate the original Procgen environments in our C-Procgen framework.
>
> ---
> **Q6:**
>
> > Reviwer: Have the authors investigated whether previous work has identified higher loss values in challenging states or environments?
>
> We are aware of methods such as PLR and PER (Prioritized Experience Replay) where higher loss samples are leveraged to accelerate training.
> However, we may not have encountered specific studies explicitly identifying higher loss values in challenging states or environments.
> We would appreciate any references the reviewer could provide to enrich our understanding and literature review.
>
> ---
> **Limitation**
>
> Thanks to reviewer for highlighting this limitation regarding the generalizability of our findings beyond PPO and their applicability to environments other than Procgen.We agree that exploring the effectiveness of our observations with value-based algorithms and in other environments would be beneficial.
> However, as our paper focuses on observations and discoveries within the C-Procgen environment, this is our primary area of emphasis.
>
> ---
> Once again, we thank the reviewer-2ABJ for his/her detailed feedback. We will address the mentioned limitations, clarities, and omissions in our citations thoroughly in our revision.

---

> > ### Comment · Reviewer_2ABJ · 2024-08-13
> >
> > Thank you for responding to my review, and I apologize for the delay in my response. I hope that the authors will address all of the writing and clarity issues by implementing the changes they have suggested here in future revisions to their paper. Aside from writing concerns, I'm not convinced by the arguments in response to W6, W7, and Q6. I suspect that these findings have been recorded in literature studying loss scales in RL because it is a commonly used intuition for methods such as PLR and PER as the authors points out. I do appreciate the contribution of recording both of these observations in Procgen, and the perspective that it forms a curriculum. However, I accounted for that contribution and assumed writing issues would be addressed when giving my original rating, so I will keep my score as is.

---

> > > ### Author Response · Authors · 2024-08-14
> > >
> > > Dear Reviewer 2ABJ,
> > >
> > > Thank you for providing further feedback! We will make sure all writing and clarity issues will be addressed in the revised version by implementing the suggested changes. We will also include a detailed discussion on how our findings connect to existing results and commonly used methods inspired by this intuition (e.g., PLR, PER).
> > >
> > > Once again, thank you for your constructive feedback. We are happy to hear you appreciate our contributions, and we will ensure the final version incorporates all the recommended revisions.

---

> ### Author Response · Authors · 2024-08-12
>
> Dear Reviewer 2ABJ,
>
> We hope you are well. We tried to consider and address the points you raised in your review, including literature comparison, clarity, and the evaluation metrics. If there’s anything further you’d like us to clarify or discuss, we would greatly appreciate your feedback.
>
> Thank you again for your time and thoughtful review!

---

### Official Review · Reviewer_7pFG · 2024-06-28

**Soundness:** 2
**Presentation:** 1
**Contribution:** 3
**Rating:** 7
**Confidence:** 4

**Summary:**

After rebuttal

I think C-Procgen is a useful contribution in itself. Secondly, the author's rebuttal has persuaded me that their analysis is novel, and can be useful. I especially like the analysis of LPE and how this relates to PLR' relative performance to PPO.

I am really impressed with the authors running my suggested LPE experiment on such short notice, and I think the preliminary results are promising.

----------------------




This paper firstly introduced C-Procgen, a version of procgen where each context variable can independently and controllably be changed, instead of the default behaviour of changing everything based on only a single seed.
Next, they analyse the distribution of contexts during training and find an implicit curriculum happens, in particular, the lengths of the episodes changes for different contexts as the agent learns. They also measure the loss per sample, and split that across contexts. They find a non-uniform split, and this also changes over training.
Finally, they investigate different procedures of changing the contexts, and the effect this has on learning.

**Strengths:**

- C-Procgen is very useful in itself. Being able to control different aspects separately is great. I would recommend releasing it as a standalone library.
- The observation that due to different episode lengths, the amount of data from different contexts changes over time as the agent learns, is interesting.
- Similarly, the correlation between PLR's performance improvement and LPE is also intriguing.

**Weaknesses:**

- While the results are interesting at a glance, they do not seem to provide any insights that could be used to develop better curriculum methods.
- The analysis, and I think conclusions, are limited to procgen, and possibly similar games. For instance, if we have a domain where all episodes have an equal length, then the different samples per context results would not hold.
- The LPE/LSE seemed somewhat tacked on, can this metric be used in some way by a method/designer?
- It it still somewhat unclear what the loss per sample is defined as, and an equation would help. Secondly, is it all the losses (e.g. policy and value) or just some?
- There are many figures, for all of the games, and this makes getting an overview of the results somewhat challenging.

**Questions:**

- what does score improvement mean in fig 4?
- Line 212-216, should it be LPE? If not, what does LSE mean?
- line 224, should that not be section 3?
- The algorithm introduced by [9] is called Prioritised Level Replay (or PLR) and not just Level Replay.
- Figure 5 has a lot of information and is somewhat hard to parse. I think it could be made clearer by having some summary plots (e.g. average test reward) as a function of the training contexts used.

**Limitations:**

- There are no suggestions or (in my opinion) useful takeaways from the analysis.

---

> ### Author Rebuttal · Authors · 2024-08-07
>
> We value the detailed feedback from the reviewer-7pFG and address each point in detail below.
>
> ---
> **W1. Insights on Curriculum Design & Limitation**
>
> Thanks for the reviewer-7pFG's feedback! While at first glance the implications for curriculum development might not be immediately apparent, our study does provide several insights that can be leveraged to enhance curriculum methods:
>
> 1. **Dynamic Adjustment of Context Sample Distribution:**
> As detailed in Section 3.1, by analyzing the learning dynamics across different contexts, we observed variations in the 'Loss Per Sample' —- an indicator of how efficiently an agent assimilates signals from its environment.
> These variations suggest that agents learn different amounts of information from different contexts at different stages of training.
> However, the distribution of context samples mostly focuses on the contexts generating lower loss values, which may not be the most informative for learning.
> This insight supports the idea of dynamically adjusting the context sample distribution based on the loss signal distribution to make learning more efficient.
> 1. **Influence of Context Gaps on Curriculum Learning:** As discussed in Section 4, the gaps between contexts significantly impact the effectiveness of the curriculum. Section 4.1 shows that missing certain contexts can hinder an agent's learning progress, while Section 4.2 demonstrates that adding new contexts can facilitate faster transfer learning to new situations.
> It implies that a well-designed curriculum should consider the distribution of context samples and the gaps between contexts to optimize learning efficiency.
>
> These insights suggest that tailored curriculum designs should not only focus on adjusting to the agent's learning needs but also optimize the distribution and progression of learning contexts to improve overall training effectiveness.
> This is precisely the purpose for which our C-Procgen was proposed: to design a system that meets these needs.
>
> ---
> **W2.**
>
> > Reviewer: The analysis, and I think conclusions, are limited to procgen, and possibly similar games. For instance, if we have a domain where all episodes have an equal length, then the different samples per context results would not hold.
>
> Thanks to the reviewer-7pFG for his/her insightful comments. Indeed, our experiments are based exclusively on C-Procgen, as the development and analysis of this new platform we introduced constitute the primary objective of our work.
>
> Additionally, it is true that in domains where all episodes have equal length, the distribution of samples per context would naturally be uniform.
> However, this does not conflict with our findings: the distribution of loss signals across different contexts does not align with the distribution of samples, which contributes to inefficiencies in curriculum learning.
> This misalignment is particularly noteworthy because the distribution of loss signals is dynamic, not fixed or uniformly distributed.
>
> ---
> **W3.**
>
> > Reviewer: The LPE/LSE seemed somewhat tacked on, can this metric be used in some way by a method/designer?
>
> As a post hoc evaluation metric, LPE (Loss Production Efficiency) cannot be directly used by designers or during method development.
>
> However, LPE can be utilized to evaluate the efficiency of loss signal production of a whole learning process, providing insights into how effectively an agent learns from its environment.
>
> In practice, rather than using LPE directly to adjust curriculum strategies, methods can leverage monitoring of the Loss per Sample to dynamically adjust the distribution of contexts within the curriculum.
>
> ---
> **W4. Clarity of Definition & Q1**
>
> Thanks for the reviewer-7pFG's valuable feedback! We will enhance the clarity in our revision regarding the definitions and calculations pertinent to our methodology.
>
> Regarding to score improvement, it is utilized as a metric to quantify the performance enhancement that the PLR (Prioritized Level Replay) method brings over the standard PPO.
> The calculation of score improvement is defined as follows:
> $\frac{score_{\text{PLR}} - score_{\text{PPO}}}{score_{\text{PPO}}}$.
> Here, $score_{\text{PLR}}$ and $score_{\text{PPO}}$ represent the scores achieved by the PLR and PPO algorithms respectively, after training for 25M steps under the same configuration.
>
> Additionally, the 'loss' referred to in the paper is the PPO Loss, because all our experiments are based on the PPO.
> The PPO Loss is in a weighted sum of three terms:
> $l_{policy}$,
> $l_{value}$,
> and $l_{entropy}$,
> and it is this weighted sum that we record.
>
> ---
> **W5.**
>
> > Reviewer: There are many figures, for all of the games, and this makes getting an overview of the results somewhat challenging.
>
> Thanks to the reviewer-7pFG for his/her feedback regarding the number of figures in our submission.
> We acknowledge that the abundance of figures for each game might complicate obtaining a clear overview of the results.
> To address this, we will attempt to use broader metrics such as Loss Production Efficiency to summarize key phenomena more succinctly in our revision.
> Given the significant diversity in game design and context spaces within Procgen, we initially included numerous figures to substantiate our points comprehensively.
>
> ---
> **Q2.**
>
> > Reviewer: Line 212-216, should it be LPE? If not, what does LSE mean?
>
> We acknowledge this and appreciate the reviewer-7pFG for pointing it out.
> The LSE at line 212-216 should indeed be LPE, which means the Loss Production Efficiency introduced at [*line 201*].
>
> ---
> **Q3 & Q4.**
>
> Thanks to the reviewer-7pFG for catching these typos. We will correct them in our revision!
>
> ---
>
>
> **Q5. Clarity of Figure5**
>
> We appreciate the reviewer-7pFG's suggestion to improve the clarity of Figure 5.
> The primary purpose of using detailed figures was to visually highlight which contexts within the environments were masked, a detail that may be crucial for nuanced analysis across different games.
>
>
> ---

---

> > ### Comment · Reviewer_7pFG · 2024-08-08
> > **Discussion**
> >
> > Thank you for your response.
> >
> > I am still not quite convinced that this work provides clear contribution besides the C-Procgen benchmark.
> > Could you please succinctly list the useful contributions of this project, and argue for how/why they matter?

---

> > > ### Author Response · Authors · 2024-08-11
> > >
> > > Dear Reviewer 7pFG,
> > >
> > > We appreciate your thoughtful comments and the time you’ve taken to review our submission. We hope our response adequately addresses your concerns, particularly regarding the detailed contributions of our work. Please feel free to share any additional feedback or suggestions, and we would be happy to address any further comments.
> > >
> > > Thank you once again for your kind consideration!

---

> > > > ### Comment · Reviewer_7pFG · 2024-08-13
> > > >
> > > > Ok, thanks again for your discussion and answers.
> > > >
> > > > I will update my score, provided that:
> > > > - All of the minor points be resolved (and that from other reviewers, e.g. 2ABJ)
> > > > - And these clarifying discussions be included in the revised manuscript.
> > > >
> > > > I've come to like figure 3 a lot, but I think it can be improved if you explain the intuition behind LPE more  in section 3.2. I do like seeing that when the LPE is high, a curriculum method like PLR doesn't do much better than PPO, since the context distribution already somewhat generates optimal losses. So please add some more explanation there.
> > > >
> > > >
> > > >
> > > > Then, relatedly. I think it would be very nice if you could run an experiment like the following:
> > > > - At every iteration, compute the LPE but do not update the agent using these losses.
> > > > - Instead, train the agent only on the top N contexts (where N can e.g. be chosen such that the new LPE is >= 80% or something). Keep the number of timesteps the same, just train on fewer (but more effective) contexts, and then update the agent using these losses.
> > > > - Compare this against standard PPO.
> > > >
> > > > I think this would help demonstrate the usefulness of LPE more, by seeing if there is a causal effect.
> > > >
> > > >
> > > >
> > > > This last experiment is not required for me to increase my score, but it would be really nice if you could add it for the camera ready version.

---

> > > > > ### Author Response · Authors · 2024-08-14
> > > > >
> > > > > Dear Reviewer 7pFG,
> > > > >
> > > > >
> > > > > Thank you for your time evaluating our work and for providing constructive and insightful suggestions.
> > > > >
> > > > > We will ensure that all minor points are addressed, including those raised by the other reviewers. We promise these clarifying discussions will be fully incorporated into our revised manuscript.
> > > > >
> > > > > Regarding Section 3.2, we will add more explanation to clarify our intuition behind the LPE metric.
> > > > >
> > > > > The experiment design you suggested is an excellent proposal; it would provide an intuitive demonstration of the LPE metric's usefulness. We started the experiments based on your suggestion, and we will share the preliminary results before the discussion period closes. We will also include the results in the final version.
> > > > >
> > > > > Thank you again for your valuable feedback and suggestions. Your insights have been incredibly helpful in refining our work.

---

> > > > > ### Author Response · Authors · 2024-08-14
> > > > >
> > > > > Dear Reviewer 7pFG2,
> > > > >
> > > > > We woud like to share the preliminary results of the experiments you suggested. These initial findings suggest that higher LPE learning processes enhance the effectiveness of agent learning.
> > > > >
> > > > > Due to time limit, we conducted the experiments only on the `Leaper` environment. We tested the standard PPO algorithm against a simple High-LPE prioritization strategy [1]. Each setting was run with three different seeds and the averaged results are reported.
> > > > >
> > > > >
> > > > >
> > > > >
> > > > > |                    | Standard PPO | High-LPE Strategy |
> > > > > | ------------------ | ------------ | ----------------- |
> > > > > | LPE                | 42.67%       | 72.53%            |
> > > > > | Avg Return         | 6.89         | 8.82 (+28.1%)     |
> > > > >
> > > > >
> > > > >
> > > > > As shown in the table[1], the High-LPE strategy resulted in a 28.1% improvement in performance, suggesting that higher LPE may positively impact learning dynamics.
> > > > > We plan to expand these experiments across all environments and run additional trials to ensure the stability and reliability of our results.
> > > > > Additionally, we will explore more High-LPE strategies to identify the most effective methods for enhancing learning efficiency.
> > > > >
> > > > > We will include more experimental results and analyses in the final revision to fully showcase our findings.
> > > > >
> > > > > Thank you for your valuable suggestions and insights!
> > > > >
> > > > > > [1]
> > > > > > Naive High-LPE strategy: In the High-LPE strategy, the agent interacts with two groups of environments: **reference environments** with a uniform context distribution and **interaction environments** where the contexts are dynamically adjusted based on the loss obtained from the reference environments. Only the loss from the interaction environments is used for backpropagation. To increase LPE, we adjust the context distribution based on the loss-per-sample metrics, which are recalculated every 5 iterations. Since the Leaper environment has a small contextual space (only 4 contexts), we did not apply the Top-N context strategy you suggested. Instead, we applied a soft strategy, sampling from the adjusted context distribution.

---

> > > > > > ### Comment · Reviewer_7pFG · 2024-08-14
> > > > > >
> > > > > > I am really impressed with you running this on such short notice (sorry!) I believe the preliminary results are promising, and I am excited to see the results for more games in the updated paper.
> > > > > >
> > > > > > I've updated my score again.

---

> > > > > > > ### Author Response · Authors · 2024-08-14
> > > > > > >
> > > > > > > Dear Reviewer 7pFG,
> > > > > > >
> > > > > > > Thank you so much for your kind words and for updating your score. We truly appreciate your encouragement and are motivated to continue refining our work.
> > > > > > >
> > > > > > > Thank you again for your support!

---

> ### Author Response · Authors · 2024-08-09
> **Response to Reviewer 7pFG**
>
> We sincerely thank the reviewer-7pFG for his/her/their prompt response! We believe our work primarily makes contributions to the field in two aspects:
>
> 1. We have introduced C-Procgen, which incorporates functionalities that enrich the community’s research tools, enabling more dynamic and versatile experiments in reinforcement learning.
> 2. Leveraging C-Procgen, we have identified an implicit curriculum in learning dynamics — a concept previously unexplored — that could inspire future studies in curriculum learning.
>
>
> ---
> **C-Procgen**
>
> - Curriculum and Transfer Learning: C-Procgen aims to enhances curriculum learning by allowing for dynamic adjustments to the learning context, helping to gauge and improve agent adaptability in varied scenarios.
> - Context-aware Learning: C-Procgen facilitates context-aware reinforcement learning, where agents adjust their strategies based on the current context.
> - Diversity and Edge Cases: C-Procgen contributes to the diversity of training environments and supports the creation of edge cases. These are vital for assessing agent resilience in non-standard conditions.
> - Environment Design: C-Procgen enables precise manipulation of game mechanics and environment attributes to study their influence on agent behavior and to experiment with new game dynamics.
>
> ---
> **Implicit Curriculum**
>
> With the help of C-Procgen, we have conducted extensive experiments to analyze learning dynamics within Procgen games.
> Our findings reveal the presence of an implicit curriculum, where despite a uniform distribution of game contexts, the learning progression of agents is notably non-uniform.
> Interestingly, agents tend to learn games in a specific order, favoring those that are neither too easy nor too challenging.
> Furthermore, the dynamics we observed, including the loss signals, the distribution of samples, and the performance of agents across different contexts, all exhibit notable interrelationships.
>
>
> This insight diverges from previous works[1][2] that primarily focus on explicitly crafting curricula to guide agents from simpler to more complex tasks. Instead, our observations suggest that the learning process itself naturally conforms to a curriculum structure — a phenomenon that has not been discussed previously. This novel perspective enhances our understanding of how agents adapt and learn, opening a new venue for future research.
>
> ---
> We appreciate the reviewer's constructive and insightful suggestions. We are happy to address any further questions and continue the discussion.
>
>
> > [1] Schaul, T., Quan, J., Antonoglou, I., & Silver, D. (2015). *Prioritized experience replay*. arXiv preprint arXiv:1511.05952.
> >
> > [2] Dennis, M., Jaques, N., Vinitsky, E., Bayen, A., Russell, S., Critch, A., & Levine, S. (2020). *Emergent complexity and zero-shot transfer via unsupervised environment design*. Advances in neural information processing systems, 33, 13049-13061.

---

### Official Review · Reviewer_h4d8 · 2024-07-13

**Soundness:** 2
**Presentation:** 3
**Contribution:** 2
**Rating:** 6
**Confidence:** 4

**Summary:**

This paper presents a benchmark called C-Procgen that builds on the existing Procgen benchmark by allowing access and control of the context parameters. Furthermore, this work investigates how learning progresses for an RL agent in the absence of a curriculum given a uniform distribution over levels. The experiments demonstrate that throughout the training the agent gets better at certain contexts first, called easy contexts with high scores and losses early on, and later on it progresses in others, i.e., hard contexts, where low scores and losses occur initially. The authors consider such progress as an implicit curriculum. In addition, their evaluation of the Level Replay algorithm in C-Procgen indicates that performance gains occur in environments with lower loss production efficiency, an effectiveness measure they introduce. Finally, they investigate the effects of partially masking and expanding training contexts, and show that some contexts, especially the ones that present moderate difficulty, are more critical to the learning progress than the others.

**Strengths:**

- The proposed benchmark provides a low degree of flexibility to research in the curriculum learning literature; hence, it is a very worthwhile contribution in itself.

- The illustrations presented in the paper depict an easy-to-understand picture of how an RL agent learns in the contextualized Procgen environment with a uniform distribution over configurations.

- Section 4 investigates masking or expanding training contexts and thus showcases interesting and valuable results for curriculum learning researchers.

- The paper is well-written and makes a significant effort to describe the implicit curriculum phenomenon.

**Weaknesses:**

- Most of the observations on learning dynamics shared in Section 3 may be because the studied environments can terminate before reaching the maximum number of steps in an episode. It would be more informative to consider the termination conditions, the reward functions, and the dynamics of these environments when evaluating how the implicit curriculum progresses.

- As most of the studied environments allow for early termination, I wonder if this causes the misinterpretation that there is an implicit curriculum. In essence, there is no curriculum being formed by the agent, but the agent progresses in certain contexts before the others, which is a valuable insight. Nevertheless, it may be misleading to say that the agent's focus shifts during training without a curriculum, as done by the authors in Section 3.2.

- Section 3.2 focuses on the Level Replay algorithm and its performance in C-Procgen environments, but this study does not investigate curriculum learning approaches such as [1,2].

[1] Klink, P., Yang, H., D’Eramo, C., Peters, J., & Pajarinen, J. (2022, June). Curriculum reinforcement learning via constrained optimal transport. In International Conference on Machine Learning (pp. 11341-11358). PMLR.

[2] Huang, P., Xu, M., Zhu, J., Shi, L., Fang, F., & Zhao, D. (2022). Curriculum reinforcement learning using optimal transport via gradual domain adaptation. Advances in Neural Information Processing Systems, 35, 10656-10670.

**Questions:**

- Are the ranges of score, loss per sample, and number of samples consistent/fixed among training stages (Figures 2 and 3) and settings (Figure 5)?

**Limitations:**

-  Please, check the weaknesses section for the comment about the lack of investigation of curriculum learning approaches developed for contextual settings.

---

> ### Author Rebuttal · Authors · 2024-08-07
>
> We truly appreciate the constructive feedback from the reviewer-h4d8 and will respond to the points raised as follows.
>
> ---
>
> **W1. Termination Conditions**
>
> We appreciate the insightful feedback from reviewer-h4d8.
> While termination conditions are indeed an important aspect to consider, they do not detract from the observations reported in our study.
>
> 1. As noted, certain environments in our study indeed terminate before reaching the maximum number of steps per episode. This premature termination can shift the sample distribution across different contexts from what might be expected under uniform distribution conditions. However, as highlighted, this does not alter our finding that the sample distribution does not align with the distribution of loss signals per sample in each context, which may lead to inefficiencies in curriculum learning.
>
> 2. Additionally, we concur with reviewer-h4d8 that incorporating a deeper analysis of termination conditions would enrich the information provided in Section 3! In our revision, we would love to also expand our discussion to include more details on the reward functions and dynamics of these environments, thereby providing a more comprehensive evaluation of how the implicit curriculum progresses.
>
>
> ---
>
> **W2. Interpretation of Implicit Curriculum**
>
> Thanks to reviewer-h4d8 for the thorough review and the insightful queries raised.
> It appears that our phrasing may have led to a misunderstanding regarding the interpretation of the 'implicit curriculum' and the intent of Section 3.2.
>
> 1. Firstly, regarding the interpretation of the 'implicit curriculum', we believe that the combination of different game contexts generated by various `level` inherently forms an implicit curriculum.
> This does not necessarily conflict with early termination of episodes.
> As reviewer-h4d8 noted, the agent does not deliberately form a curriculum from the environmental contexts; however, for the agents, these varying contexts of difficulty inherently serve as a curriculum, guiding their learning progression.
>
> 2. Additionally, I would like to clarify a point regarding the agent's focus during training. Our discussion in the manuscript seems not assert that '*the agent's focus shifts during training without a curriculum*'.
> Instead, we examine how the distribution of loss signals and samples across contexts influences the learning process.
> We introduce the concept of Loss Production Efficiency to assess this dynamic.
> This metric helps us understand the effectiveness of a curriculum and the agent's interaction with it.
>
> We appreciate the opportunity to clarify these points and will ensure that our revision reflects these clarifications to avoid any potential misinterpretations.
>
>
> ---
>
> **W3. Evaluation Metrics**
>
> Thank you for your observation regarding Section 3.2, which indeed focuses on utilizing the Prioritized Level Replay (PLR) to illustrate the mismatch between loss and sample distribution in the C-Procgen environments.
>
> We are grateful to the reviewer for bringing to our attention the studies by Klink et al. (2022) and Huang et al. (2022).
> Upon a thorough review of these works, we agree that they offer valuable insights and present methodologies that are excellent and relevant to our field of study.
> These studies provide advanced approaches to curriculum reinforcement learning, which could enrich our understanding and application of similar concepts in future research.
>
> However, due to the significant variability across games in Procgen, replicating similar presentations to demonstrate the learning dynamics of agents under a curriculum poses challenges.
> Each game's unique characteristics and context spaces necessitate a tailored approach when applying curriculum learning strategies, which our current focus on PLR aims to address in context-specific manners.
>
> ---
>
> **Q1. Consistency of Metrics**
>
> In our Figures 2, 3, and 5, the range of the score metric is consistent across different training stages and settings.
> For other metrics, they are normalized on each heatmap for clarity and visibility.
> Comparing these metrics, especially 'loss per sample', at different times can be challenging due to their significant scale variability during training.
> We recognize the importance of this clarification and will address this explicitly in the versions to ensure the data's context and limitations are well understood.

---

> > ### Comment · Reviewer_h4d8 · 2024-08-12
> > **Re: Rebuttal by Authors**
> >
> > Thank you for responding to my comments and questions. I'd like to apologize for the delay caused by travel.
> >
> > ** W1. Termination conditions: **
> >
> > > This premature termination can shift the sample distribution across different contexts from what might be expected under uniform distribution conditions.
> >
> > Although I agree with the rest of the first section of the response, I'm afraid I have to disagree with this statement. As far as I understand, there is no distribution shift during training if we talk about distributions over the task space.
> >
> > ** W2. Interpretation of Implicit Curriculum:**
> >
> > Thank you for the clarification. Please reflect on this in the manuscript as well.
> >
> > ** W3. Evaluation metrics. **
> >
> > > However, due to the significant variability across games in Procgen, replicating similar presentations to demonstrate the learning dynamics of agents under a curriculum poses challenges.
> >
> > Such a perspective would hurt the applicability of C-Procgen, which would be very useful for research in curriculum learning for contextual RL.
> >
> > PLR-like approaches are developed primarily for Procgen environments, where contexts are not accessible, and each context/task/level is labeled with an ID. Although C-Procgen can bring a lot of insights into PLR-like method design, its fundamental impact would be on curriculum learning for contextual RL, which you talk about in the second subsection of the related work.
> >
> > ** Q1. Consistency of Metrics**:
> >
> > Thank you for the clarification.
> >
> > I appreciate the effort put into the rebuttal and the new figures in the shared PDF. I will stick with the original score I gave.

---

> > > ### Author Response · Authors · 2024-08-12
> > >
> > > We highly appreciate the effort and thoughtful feedback provided by reviewer h4d8. We would love to briefly address the concerns raised and provide further clarification:
> > >
> > > ---
> > > **W1: Distribution Shift**
> > >
> > > The term "distribution shift" refers to the changes in the distribution of samples across different contexts.
> > > When the probability of each game context being initialized is equal, longer episode lengths result in more training samples coming from certain contexts.
> > > As episode lengths change, this leads to a shift in the distribution of samples across the various contexts.
> > >
> > > ---
> > > **W3. Evaluation metrics**
> > >
> > > We agree with the reviewer-h4d8 that the applicability of C-Procgen is crucial.
> > > Our C-Procgen framework offers easy and unified parameters to set up game contexts.
> > > We acknowledge that unifying the presentation of learning dynamics across different games is challenging due to their varying context spaces.
> > > We will explore new approaches to better represent these learning dynamics in future work!
> > >
> > > ---
> > >
> > > We greatly value the reviewer-h4d8’s thoughtful evaluation and the time spent reviewing our work.
> > > Thanks once again for his/her/their insightful comments and for helping us improve our research.

---

### Author Rebuttal · Authors · 2024-08-07

We would like express our sincere gratitude to all reviewers fortheir constructive comments!
We are particularly thankful for the following positive feedback:

- The proposed benchmark is a very worthwhile contribution. `Reviewer h4d8`, `Reviewer 7pFG`, `Reviewer 2ABJ`;
- The figures in the paper are easy-to-understand. `Reviewer h4d8`, `Reviewer 2ABJ`;
- The observations are very interesting. `Reviewer 7pFG`;
- The results are valuable for curriculum learning research. `Reviewer h4d8`, `Reviewer 2ABJ`;
- The methodology, presentation of results, and analyses are all sound. `Reviewer 2ABJ`;

We wil address the specific questions raised by the reviewers in the subsequent sections of the rebuttal.

Additionally, a new illustration of learning dynamics with learning curves and more detailed metrics information can be found in the PDF file.

---

> ### Author Response · Authors · 2024-08-14
> **Acknowledgment**
>
> Dear Reviewers and Area Chairs,
>
> We sincerely thank the reviewers and area chairs for the time and effort they put into reviewing our submission. The reviewers' thoughtful feedback and suggestions have been especially valuable in improving our work.
>
> We deeply appreciate the reviewers' insights and the guidance provided. Thank you once again to everyone for their support!

---

### Decision · Program_Chairs · 2024-09-25

**Decision:**

Accept (spotlight)

**Comment:**

This is in interesting analysis paper, which investigates the implicit curriculum in procgen that results simply from the agent's learning. The authors find that just due to the learning dynamics there is a shift for easy contexts to harder context, i.e. an implicit curriculum.

The reviewers are in agreement that this analysis is useful and that even the new environment (c-procgen) is a useful contribution to the community.

I therefor advocate for acceptance as a spotlight at the conference.